# Structural and Electrochemical Behaviors of ZnO Structure: Effect of Different Zinc Precursor Molarity

**Ruziana Mohamed [1],\* and Muhammad Syakir Azri Anuar [2]**

[1] Faculty of Applied Sciences, Universiti Teknologi MARA (UiTM), Shah Alam 40450, Selangor, Malaysia
[2] Faculty of Applied Sciences, Universiti Teknologi MARA Pahang,
Bandar Tun Razak Jengka 26400, Pahang, Malaysia
\* Correspondence: ruzianamohd@uitm.edu.my or ruziana12@gmail.com

**Abstract:** This research synthesised zinc oxide (ZnO) structure by a hydrothermal method. ZnO samples were prepared using different molarities of zinc (Zn) precursor, ranging from 0.10 to 0.16 M. Structural and morphological properties were characterised by X-ray diffraction (XRD) and scanning electron microscopy (SEM). The XRD patterns show that all samples are prominently grown along the three diffraction peaks at (001), (002) and (101) planes. The ZnO sample with 0.16 M Zn precursor has the highest peak orientation along the (002) plane. The average crystallite sizes for the ZnO structure with 0.10, 0.12, 0.14 and 0.16 M precursor are 48, 51, 49 and 31 nm, respectively. ZnO sample prepared at 0.16 M has the smallest crystallite size and the lowest tensile strain. The SEM images show that the ZnO samples are randomly oriented with average diameters of 209, 325, 295 and 348 nm when using 0.10, 0.12, 0.14 and 0.16 M of the precursor, respectively. The electrochemical behaviour of the ZnO structure was determined through cyclic voltammetry (CV) measurement. In the CV curve, the calculated specific capacitance for the ZnO sample prepared at 0.16 M has the highest value of $3.87 \, \text{Fg}^{-1}$. The ZnO sample prepared at 0.10 M has the lowest specific capacitance value of $2.11 \, \text{Fg}^{-1}$. Therefore, changing the molarity of the Zn precursor could change the structural and electrochemical properties. ZnO sample prepared with 0.16 M of the precursor provides the optimal result.

**Keywords:** zinc oxide precursor; hydrothermal; structural properties; electrochemical behaviour

## 1. Introduction

Nanotechnology and nanoscience are closely linked fields that involve altering matter at the atomic or molecular level to produce new products and technologies with distinctive qualities. These fields have gained increased attention due to its capacity to radically alter the physical, chemical and optical properties of metals, metal oxides and metalloids by shrinking them to the nanoscale [1]. Zinc oxide (ZnO), a metal oxide, is one of the most well-known materials due to its broad band gap, good chemical and thermal stability and optoelectronic and electrical capabilities when the material is fabricated into nanostructures [2]. ZnO is an ideal material for applications, such as transparent electronics, ultraviolet light emitters, piezoelectric devices, transistors, solar cells, batteries, supercapacitors, and catalysts [3]. It is also used to fabricate biosensors and chemical sensors that show good sensitivity due to their high surface area-to-volume (S/V) ratio [4].

Numerous studies have been conducted to control the development ZnO nanostructures for specific applications by adjusting the synthesis process, reactant and parameter [5–7]. In the literature, Perillo et al. (2017) fabricated ZnO structure by using a chemical method at low temperatures. The order of adding reactants was varied, and the structures were classified into two types, i.e., Type 1 and Type 2. For Type 1, zinc acetate dihydrate was dropped into sodium hydroxide (NaOH); for Type 2, the order of adding reactants was the opposite of that for Type 1. The Type 1 ZnO sample had hexagonal-shaped nanorods with a specific surface area larger than the Type 2 ZnO sample that had flower-shaped nanorods [8].

Zhang et al. (2021) successfully produced ZnO quantum dots (QDs) through sol–gel method by varying three different alkali bases, i.e., lithium hydroxide (LiOH), potassium hydroxide (KOH) and NaOH. They found that smaller particle sizes could be produced efficiently at a higher OH concentration; however, a higher OH concentration negatively affected water solubility and fluorescence intensity. ZnO QDs synthesised with LiOH and NaOH as basic ingredients had smaller particle sizes than those made with KOH [9].

Jayaraman et al. (2018) synthesised ZnO thin film through spray pyrolysis using different types of precursors. The zinc precursors used are zinc acetylacetonate, zinc chloride and zinc acetate. The crystallite sizes of the ZnO thin film range from 21 nm to 59 nm. Various shapes of the ZnO structure were obtained: hexagonal layers, hexagonal nanorods and nanothorns in nanoscales with elongated grains. Hexagonal plates and nanothorns were randomly arranged in the film, exposing a wide range of ZnO facets. However, when the precursor concentrations increased, these nanothorns combined into bigger structures, thereby decreasing the number of exposed faces [10].

Amirthavalli et al. (2018) synthesised ZnO NPs by varying the concentration of zinc (Zn) precursor with the assistance of CTAB surfactant prepared at room temperature. The ZnO structure had crystallite sizes and pore diameters between 20 and 40 nm and between 20 and 33 nm, respectively. At higher concentrations of Zn acetate, the morphology of the particles changed from edge-rounded triangular-shaped to rough-edged quadrilateral-shaped particles [11].

Singh et al. (2009) determined the effects of precursor concentration solutions on ZnO growth by using spray pyrolysis and varying the molarity values of zinc nitrate solution from 0.01 to 0.2 M. They discovered that the crystalline quality as well as the microstructure of films were influenced by solution molarity. When the molarity of the precursor solution increased, the grain size and roughness of the ZnO sample increased, followed by a decrease in resistivity. Similarly, when the molarity of zinc nitrate increased, the activation energy decreased, thereby increasing the carrier concentration [12].

Based on literature, the properties of the ZnO structure can be changed by varying precursor type, precursor molarity, and reactive agent during preparation. Different preparation methods affect the formation of ZnO structures.

A crucial element that needs further research involves controlling the structural and morphological characteristics that may influence the surface area-to-volume ratio. However, creating an ideal structure with high surface area while controlling particle size distribution remains challenging.

Fahimi and Moradlou (2020) stated that the effectiveness of electrical and energy storage devices was constrained by ZnO low surface area as well as its active sites, leading to poor electron mobility. To improve the performance of electrical devices, we must increase the surface area to promote electron mobility [13]. Hence, by varying the molarities of zinc precursors, the desired structures can be fabricated [14].

In this study, we focused on the synthesis of ZnO structure by using a simple hydrothermal method with simple modifications. Previous research used surfactant to assist in the growth of ZnO structure with a longer time for heating. Here, we add to previous knowledge by preparing ZnO structure based on a solution method without using any surfactant, with a shorter heating time. Different types of zinc nitrate precursor were used, and the structural and electrochemical properties of ZnO structure were investigated.

## 2. Results

### 2.1. Structural Properties

Figure 1a shows the crystallinity of ZnO structures based on their XRD patterns. The diffraction peak angles are all matched with the JCPDS data (Card No. 36-1451), revealing the formation of a wurtzite phase. No other peaks exist other than the ZnO peaks, indicating the zero impurity of the sample. The sharp peaks with the strongest intensities in the (100), (002) and (101) planes for all samples indicate good crystallinity and higher grain boundary density [15,16]. Figure 1b shows the comparison among the three dominant peaks of the

samples and the diffraction angles of reference data. All the peaks of the samples are shifted towards the lower diffraction (2-theta) angle, indicating the increased interplanar distance of the ZnO crystal [17].

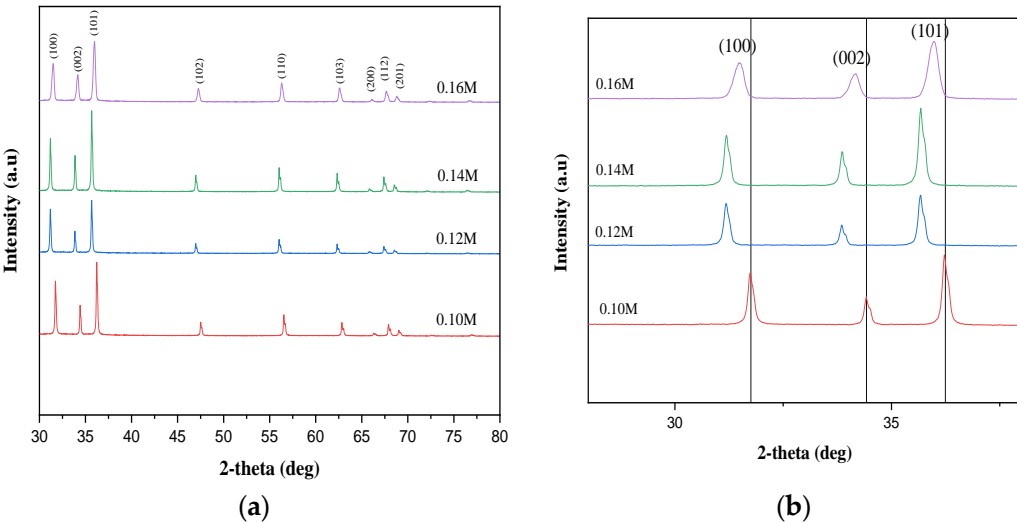

**Figure 1.** XRD peak pattern for ZnO structure prepared at different molarities of Zn precursor ranging from 0.10 to 0.16 M: (**a**) XRD pattern for all diffraction peak at 30–80°; (**b**) enlarged image for XRD pattern of three dominant peaks at the (100), (002), and (101) planes.

The relative peak intensity orientation, P(*hkl*), of the three dominant planes was calculated using Equation (1) to identify the highly oriented plane [12]:

$$\mathrm{P}_{(hkl)} = \frac{I_{(hkl)}}{\sum I_{(hkl)}} \tag{1}$$

where *I*(*hkl*) is the measured relative intensity plane, and $\Sigma I(hkl)$ is the intensity of all the diffraction peaks in all planes.

Figure 2 shows the comparison of P for the three dominant peaks of all ZnO samples. The diffraction peak at the (101) plane shows the highest intensity, indicating that the growth is prominent along the (101) plane, except for 0.12 M because the growth strengthens along the (100) plane. The samples show low (002) peak intensity due to the weakened preferred orientation along the (002) plane [18]. The ZnO sample prepared at 0.16 M zinc (Zn) precursor showed the highest peak intensity along the (002) plane compared with the other samples. Hence, this sample has the lowest surface free energy of the (002) plane, resulting in preferential growth along the c-axis orientation and enhancement in the structural order compared with the other samples [12,19].

The average crystallite sizes of all ZnO samples of the three dominant crystal peaks at the (100), (002) and (101) planes were calculated using the Debye–Scherrer Equation (2) [20,21],

$$D = \frac{0.94\lambda}{\beta \cos \theta} \tag{2}$$

where *D* is the crystallite size, $\lambda$ is the X-ray wavelength, $\beta$ is the full width at half maximum (FWHM) in radians, and $\theta$ is the Bragg's angle in degrees.

Figure 3 shows the calculated average crystallite size for all ZnO samples. The average crystallite sizes for ZnO samples prepared with 0.10, 0.12, 0.14 and 0.16 M Zn precursor are 48, 51, 49 and 31 nm, respectively. The crystallite size of the prepared samples increases with increasing Zn precursor molarity but decreases with further increase in the molarity. Variations in the molarity of the Zn precursor influence the structure of the ZnO samples, where the Zn ion concentration affects the nucleation process and growth of structures, thereby influencing the shape, size and density of the samples [22,23]. The ZnO sample



prepared at 0.16 M has a smaller crystallite size than the other samples, leading to increased availability of active sites. Previous works reported that small crystallite sizes were produced due to rapid nucleation [19]; in the present results, this rapid nucleation process occurred at higher precursor concentration. Mohamed et al. (2020) stated that a smaller crystallite size could contribute to an increase in the aspect ratio and improve the active sites of a sample [24]. According to Bui et al. (2021), a small particle size could provide a larger surface area and increase the active sites and the specific capacitance value [25].

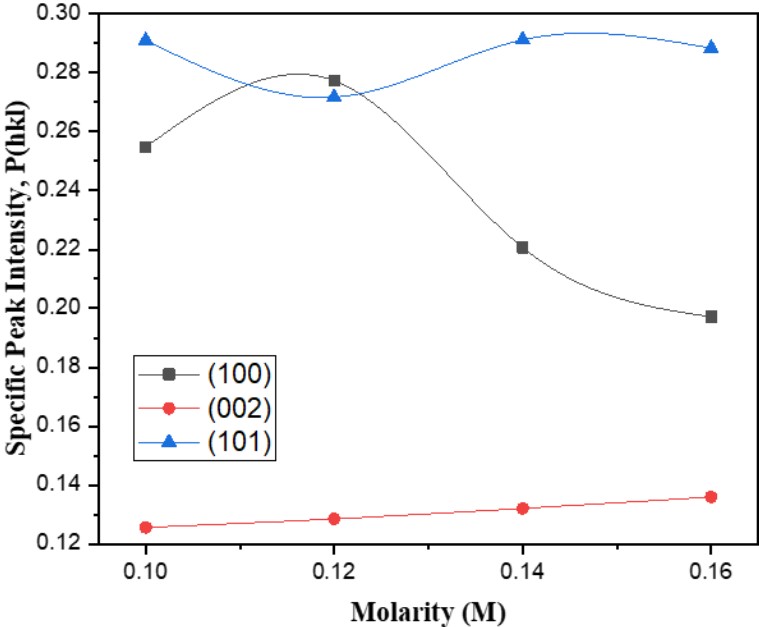

**Figure 2.** The relative peak intensity, P, for three dominant peaks of ZnO samples at different molarities.

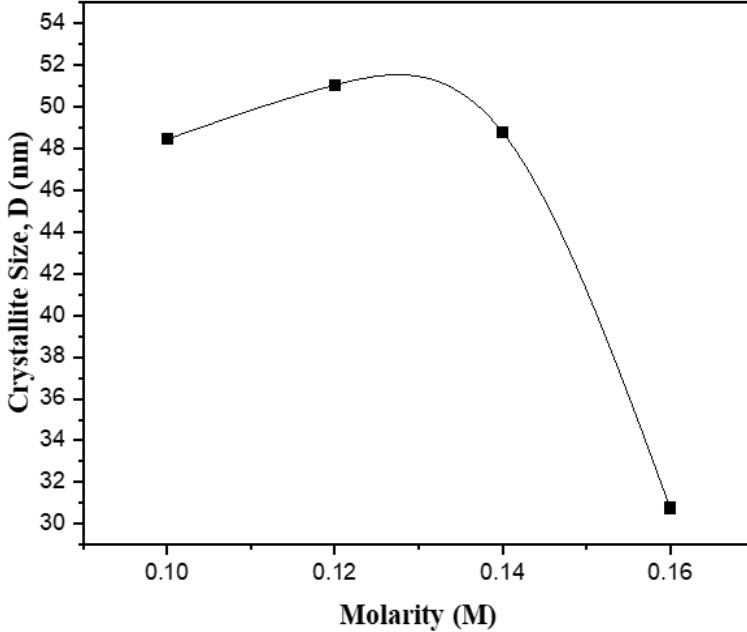

**Figure 3.** The crystallite size of ZnO structure at different Zn precursor molarities.

The lattice constants, (*a* and *c*) [26], atomic packing factor, *APF* [27], strain $\varepsilon$ and stress $\sigma$ [28] of the ZnO samples were calculated using Equations (3), (4), (5), (6) and (7), respectively, and the calculated data are shown in Table 1:

$$a = \frac{\lambda}{\sqrt{3}\sin\theta} \tag{3}$$

$$c = \frac{\lambda}{\sin\theta} \tag{4}$$

$$APF = \frac{2\pi a}{3\sqrt{3}c} \tag{5}$$

$$\varepsilon = \frac{c - c_0}{c_0} \tag{6}$$

$$\sigma = \frac{2C_{13}^2 - C_{33}(C_{11} + C_{12})}{2C_{13}} \times \varepsilon \tag{7}$$

where $\lambda = 1.54$ Å; $c_0 = 5.206$ Å is the lattice parameters of unstrained (bulk) ZnO; while $C_{11} = 209.7$ GPa, $C_{12} = 121.1$ GPa, $C_{13} = 105.1$ GPa, and $C_{33} = 210.9$ GPa are the elastic stiffness constants of bulk ZnO.

**Table 1.** Lattice parameters $a$ and $c$, ratio of $a/c$, *APF*, strain and stress values, and dislocation density of ZnO samples.

| Different Molarity of Zn Precursor (M) | Lattice Parameter, $\left(\text{Å}\right)$ | | Ratio *a/c* | *APF* (%) | Strain, $\varepsilon$ % | Stress, $\sigma$ $(\times 10^5$ **Pa**$)$ | Dislocation Density, $\delta(\text{nm})^{-2}$ |
|---|---|---|---|---|---|---|---|
| | $a(100)$ | $c(002)$ | | | | | |
| 0.10 M | 3.2522 | 5.2052 | 1.6005 | 75 | −0.015 | 0.35 | 4.26 |
| 0.12 M | 3.3075 | 5.2889 | 1.5990 | 75 | 1.59 | −36.1 | 3.84 |
| 0.14 M | 3.3075 | 5.2889 | 1.5990 | 75 | 1.59 | −36.1 | 4.20 |
| 0.16 M | 3.2779 | 5.2404 | 1.5987 | 75 | 0.66 | −14.1 | 10.56 |

The lattice parameters $a$ and $c$, ratio of $a/c$, *APF*, value of strain, and stress are summarised in Table 1. The ZnO samples all have the same *APF* value of approximately 75%, which is slightly larger than the *APF* of bulk hexagonal ZnO materials (about 74%). The negative sign of strain indicates that it is compressive which refers to lattice contraction. Meanwhile, the positive sign of strain shows that it is tensile, in which the lattice constant undergoes expansion. At lower concentrations of the Zn precursor, the ZnO sample is in compressive strain, indicating that the sample material is under compression. At higher concentrations of the precursor, the ZnO samples are in tensile strain and considered to be stretched [16,29]. The lowest tensile strain of the ZnO samples occurs when the sample is prepared with 0.16 M Zn precursor, indicating that this sample is in the relaxation state of the crystal structure [16]. The electron transport in a ZnO sample can be improved when the sample is under minimizal strain [16]. The dislocation density, $\delta$, is calculated using Equation (8) [30], where $D$ is the average crystallite size:

$$\delta = 1/D^2 \tag{8}$$

The value of $\delta$ describes the physical defect in the crystal and is inversely proportional to the square of crystallite size along the three dominant planes [29]. The dislocation density of the ZnO sample prepared at a molarity of 0.12 M has a minimum value. The maximum value occurs for the ZnO sample prepared at a molarity of 0.16 M. A small crystallite size with a large dislocation density value indicates ordered grains with higher micro defects in the grains of the ZnO sample [19].

### 2.2. Morphological Analysis

SEM analysis was performed to evaluate the morphological structure of the ZnO samples. Figure 4 shows the SEM images of the ZnO samples prepared with different Zn precursor solutions. The obtained results reveal that the concentration of the Zn precursor affects the growth of ZnO structure. From the observations, the morphology in all sample powders are rod-like structures that are randomly oriented. A compact rod structure was

observed for the ZnO sample prepared with 0.10 M of the Zn precursor. A hexagonal-shaped rod structure was obviously seen for the sample prepared with 0.12 M of the Zn precursor. The rod of the ZnO structure becomes slightly elongated when the Zn precursor concentration increases at a molarity of 0.12 M (Figure 1b). The ZnO sample starts to shorten again, forming overlapping spherical-like structures by increasing the Zn precursor concentration to 0.14 and 0.16 M. The size distributions of the ZnO structures were investigated by considering the particle sizes in the histograms in Figure 4b,d,f,h. The average diameters for the ZnO structures prepared with 0.10, 0.12, 0.14 and 0.16 M Zn precursor are 209, 325, 295 and 348 nm, respectively. The average lengths of the ZnO rod structures are 668, 1040, 372 and 471 nm in the samples prepared with 0.10, 0.12, 0.14 and 0.16 M of Zn precursor, respectively. The decreasing or increasing diameter and length of the ZnO structures are caused by diversion growth in various axes, which are influenced by the endothermic growth process [31]. The calculated aspect ratios are 3.20, 3.20, 1.26 and 1.35 for the four ZnO samples prepared with 0.10, 0.12, 0.14 and 0.16 M Zn precursor, respectively.

*2.3. FTIR Analysis*

The spectra of the FTIR analysis of ZnO structure is shown in Figure 5. The typical absorption bands in the wavenumber region 400–600 cm$^{-1}$ are the fingerprint area of metal oxygen bonds [30,32]. The existence of absorption bands in the FTIR spectra can be assigned to Zn-O and Zn-OH stretching vibrations. The FTIR spectra reveal the lack of other metal oxygen bond except the Zn-O stretching. Other absorption bands at around 900 cm$^{-1}$ are due to Zn-OH stretching vibrations [33,34]. All the samples have a higher presence of Zn-OH stretching vibrations than Zn-O stretching vibrations. The excessive OH ion from the NaOH solution might cause the presence of Zn-OH stretching.

*2.4. Cyclic Voltammetry (CV) Measurement*

The electrochemical characteristic of ZnO was tested on a glass carbon electrode (GCE) and examined using cyclic voltammetry with silver/silver chloride (Ag/AgCl) as a reference electrode and 0.1 M sulphuric acid ($H_2SO_4$) as a supporting electrolyte. Figure 6 shows the results of the CV measurements for all samples. The CV curves for all samples have a symmetric-like relation of charge and discharge parts, indicating the good electrochemical behaviour and reversibility of the redox reaction [13].

The specific capacitances, *Cs*, of the samples was calculated using Equation (9) [35]:

$$C_s = \frac{\int I\, dV}{v \times m \times \Delta V} \tag{9}$$

where $\int I\, dV$ is the total integrated area of the CV curve, $v$ is the scan rate (mVs$^{-1}$), $m$ is the mass of the electroactive material in the electrode (g), and $\Delta V$ is the sweep potential window.

Table 2 shows the data for area integrated CV and calculated *Cs* of ZnO structure at different molarities of Zn precursor. the data for area integrated CV and calculated *Cs* (3.87 Fg$^{-1}$). The ZnO structure prepared with 0.12 M Zn precursor has the smallest area integrated CV and *Cs* value of 2.03 Fg$^{-1}$. Although the aspect ratio obtained from the SEM analysis is higher for the ZnO sample prepared at 0.12 M, the crystallite size and tensile strain are larger. The low peak intensity along the (002) plane may reduce the value of *Cs*.

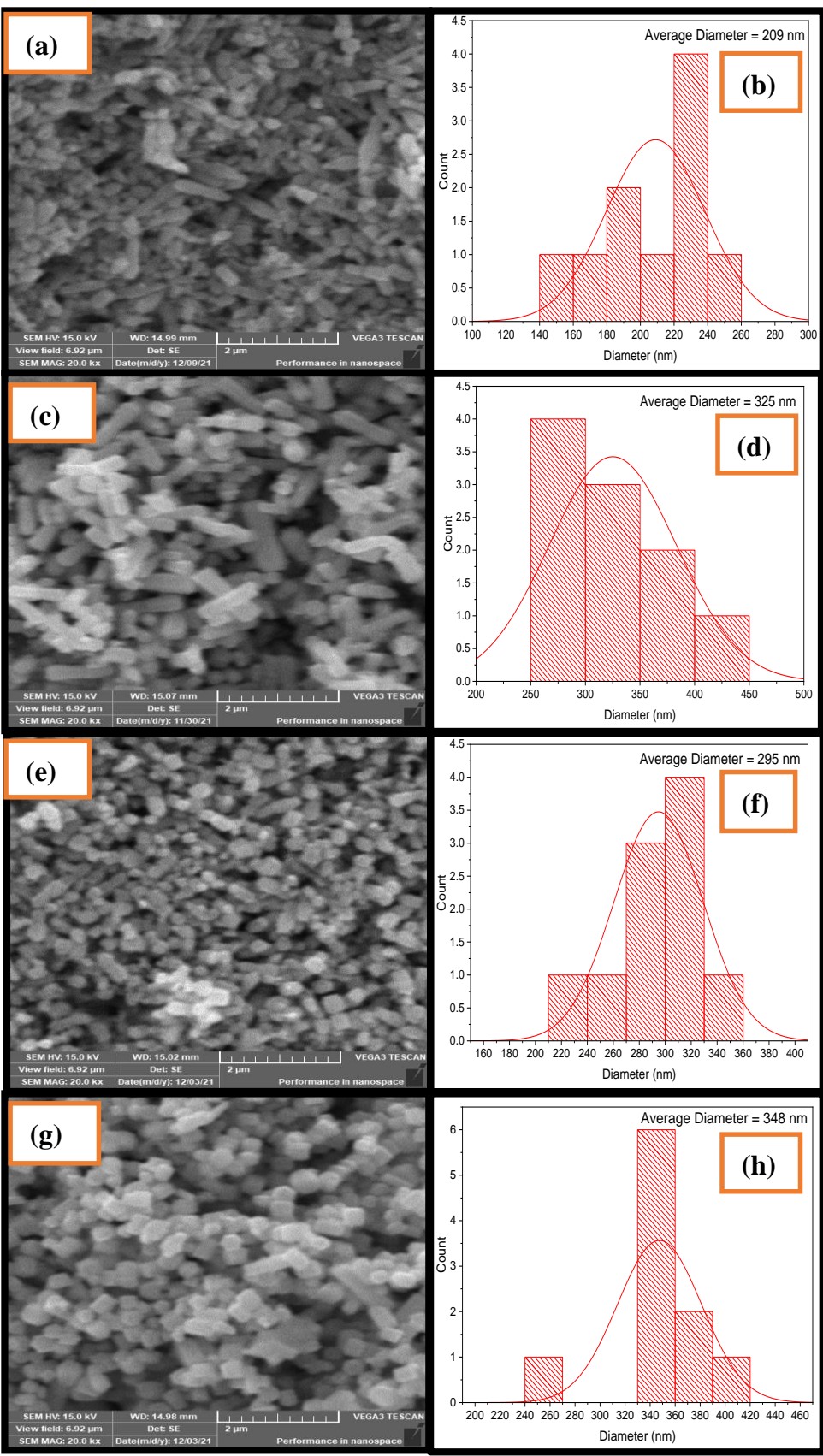

**Figure 4.** SEM images at 20k magnification for different zinc precursors concentrations: (**a**,**b**) 0.10 M; (**c**,**d**) 0.12 M; (**e**,**f**) 0.14 M; (**g**,**h**) 0.16 M.

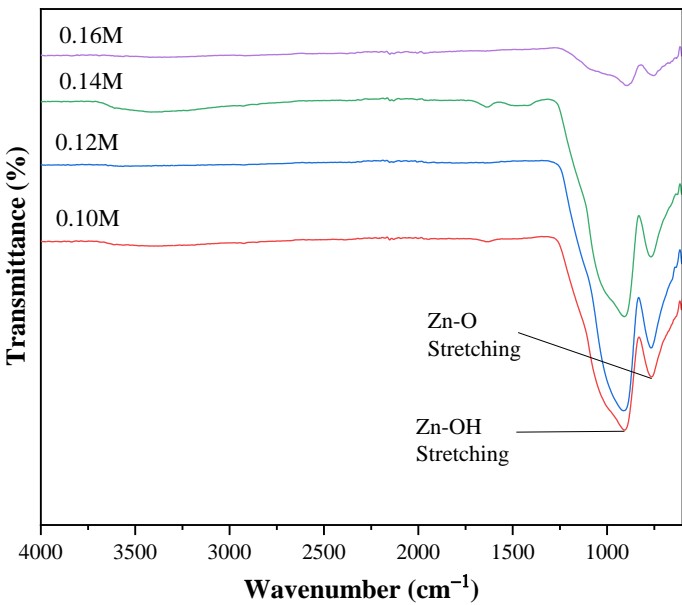

**Figure 5.** FTIR spectra of ZnO structure at different Zn precursor molarities.

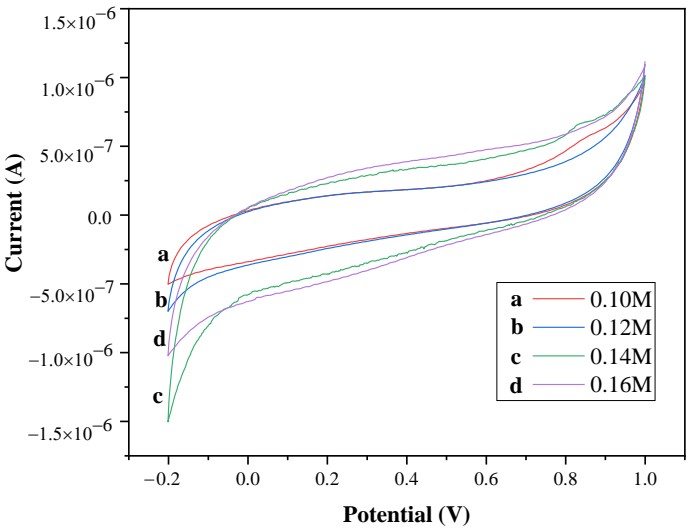

**Figure 6.** CV curves of ZnO structure samples with variations in Zn precursor molarity.

**Table 2.** Area integrated CV and specific capacitance for ZnO samples with variations in Zn precursor molarity.

| Different Molarity of Zn Precursor (M) | Area ($10^{-7}$ AV) | $Cs$ (Fg$^{-1}$) |
|:---:|:---:|:---:|
| 0.10 | 4.00 | 2.11 |
| 0.12 | 3.85 | 2.03 |
| 0.14 | 6.70 | 3.54 |
| 0.16 | 7.33 | 3.87 |

## 3. Discussion

The properties of ZnO structures change with changes in the molarity of the Zn precursor. At a high concentration of Zn precursor (0.16 M), the ZnO sample has a small crystallite size, leading to the availability of more active sites. The growth of the ZnO samples occurs at three dominant peaks at the (001), (002) and (101) planes. The XRD pattern at the (002) plane increases as the Zn precursor concentration increases, indicating the sample growth at a lower surface free energy with well-ordered grains formed. At a

higher concentration of Zn precursor, lower tensile strain occurs, indicating that this sample is in relaxation, resulting in enhanced electron transport. Xue et al. (2020) stated that a higher *Cs* value indicates that the ZnO structure has fast rate electron transfer between the active materials and that the charge collector due to the ZnO structure has more active sites inside the structure. A low *Cs* value occurs because the rate of electron transfer is slow due to the reduced active area, which makes it unbeneficial to increase the effective liquid–solid interface area, leading to the inefficient utilisation of the active material [36]. Although the diameter of ZnO structure prepared with 0.10 M Zn precursor is smaller than that of the other samples, the distribution does not seem to be good due to the observed compact structure. Future research directions should include analysis of porosity, specific area over volume (S/V) and the electrochemical stability to learn more information about the structural and electrochemical properties of ZnO structures. Tuning of the preparation process, such as prolonging the growth process at temperatures above 150 °C, may be necessary to change the structure properties and electrochemical behaviour to meet specific applications.

## 4. Materials and Methods

### 4.1. Synthesis of ZnO Structure

Zinc nitrate hexahydrate, $Zn(NO_3)_2 \cdot 6H_2O$, and sodium hydroxide (NaOH) were used as precursor and reagent, respectively. The materials were dissolved in DI water and mixed. The precursor solutions were prepared at four different molarities, i.e., 0.10, 0.12, 0.14, and 0.16 M into 0.1 M NaOH. The mixed solution was stirred for 10 min, poured into a 100 mL Teflon cylinder and placed in autoclave for hydrothermal process. The autoclave was heated at 150 °C for 8 h. The cloudy solution inside the Teflon was centrifuged, and the precipitate was washed with DI water followed by ethanol and then dried.

### 4.2. Characterization

The ZnO sample powders were characterised by X-ray diffraction (XRD) using CuK$\alpha$ X-ray ($\lambda$ = 1.54 Å) and scanning electron microscope (SEM) to investigate the structural and morphological properties. Fourier transform infrared (FTIR) through attenuated total reflection (ATR) method within wavelength range 4000–400 cm$^{-1}$ and cyclic voltammetry measurement (Autolab PGSTAT204) was performed at the potential window from −0.2 to 1.0 V and scan rate of 20 mVs$^{-1}$ to identify the chemical bonds and determine the electrochemical behaviour. The preparation and characterisation process is presented in Figure 7.

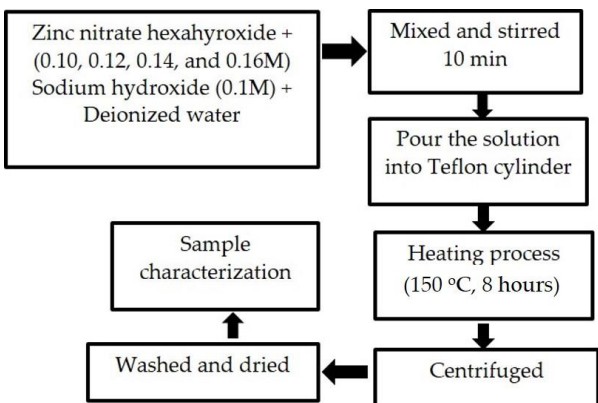

**Figure 7.** Flow chart of the preparation process for the ZnO samples.

## 5. Conclusions

ZnO structures were successfully fabricated by a simple hydrothermal method. Most of the ZnO structures are dominantly oriented along the (101) plane, followed by the (001) and (002) planes. The ZnO sample prepared at a molarity 0.16 M has the highest peak intensity along the (002) plane and the lowest tensile strain, indicating that it is under

stretched compared with the other samples. This sample also has the smallest crystallite size with well-ordered grains that contribute to the existence of more active sites. The SEM images show that ZnO structures are randomly oriented with rod shapes appearing at a lower concentration. The shapes become larger, with some shapes such as rectangular and hexagonal shapes appearing at higher concentrations of the Zn precursor. The highest *Cs* value is 3.87 Fg$^{-1}$ for the ZnO structure sample with 0.16 M Zn precursor. The high *Cs* value for the ZnO sample prepared at a molarity of 0.16 M may be due to the small crystallite size and low tensile strain of the sample. Hence, the molarity of zinc precursor affects the structure and electrochemical behaviour of the samples. The obtained results may exhibit significant capabilities in electrical storage applications.

**Author Contributions:** Conceptualization, R.M. and M.S.A.A.; investigation, M.S.A.A. and R.M.; writing—review and editing. All authors have read and agreed to the published version of the manuscript.

**Funding:** This research was funded by the Ministry of Higher Education (MOHE), research grant FRGS/1/2019/STG07/UITM/02/22.

**Informed Consent Statement:** Not applicable.

**Data Availability Statement:** Not applicable.

**Acknowledgments:** The authors would like to thank the Research Management Institute (RMI) of UiTM, Faculty of Applied Sciences, UiTM Shah Alam and UiTM Pahang for providing the facilities.

**Conflicts of Interest:** The authors declare no conflict of interest.

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
