# Peer review of "Structural and Electrochemical Behaviors of ZnO Structure: Effect of Different Zinc Precursor Molarity"

_condensedmatter, doi:10.3390/condmat7040071_

Round 1

Reviewer 1 Report

The current article not suitable for publication in Condensed matter. The major revision are necessary to publish in this journal based on comments below:

 1.                  XRD result show the crystallite size of ZnO nanostructures prepared at different concentration. Can it effect to the grain inside the structure?

2.                  Figure 1 is not clear, Please improve the quality.

3.                  Please provide the equation of stress and strain based on obtained XRD result to see the relation with the peak intensity or the calculated crystallite size. Discuss in terms of structural changes when concentration of precursor increases. 4.                  “In order to improve the performance of electrical devices, we must thus enhance the surface-area to volume ratio, which increases electron mobility [12]. Hence, by varying the molarities of zinc precursors, the desired structures can be fabricated.” This statement should be supported by references (here is are suggested ones https://doi.org/10.1016/j.ijleo.2022.169137, https://doi.org/10.1007/s10854-022-08440-1)

5.                  Will the XRD results above affect the electrochemical behaviour? Proposed to discuss the relationship between calculated stress and strain data and the electrochemical properties of the sample.

6.                  It is suggested that the author can prepare a flow chart in the experimental method to help the reader understanding.

Author Response

Dear reviewer,

Reviewer 2 Report

The authors have described relevant process development results in this manuscript. Majority of the paper is difficult to read, and the sentences need to be edited for language. Beyond this comment, here are few questions the authors should explain in a revised manuscript:

1. In Section 2.1, line 96, "More pores appear with increasing Zn precursor molarity". Kindly explain what you mean by "pores". Also, annotate these pores in the SEM images to make it easier for a reader to follow the discussion. This is also referred to later in Lines 261-262, so a clear explanation here will be needed to understand the conclusion.

2. Lines 100-104. Must explain why the average diameter and length increases only between 0.1-0.12M but then decreases at higher molarity.

3. Add a detailed procedure in the Methods section 4.2, about how the particle ZnO rod diameter and length were analyzed from the SEM images. Further, provide statistical data from this calculation, including, number of rods, standard deviation, number of spots in the sample. This will be crucial for readers interested in replicating your work.

4. The crystallite size explained in the text (lines 170-172) do not match with the graph plotted in Figure 3. The graph and the text need to be corrected. Since these sizes are within the calculation error, suggest adding a table with the numbers used for measurement, Bragg angle, FWHM, crystallite size for each case. 

5. Line 172 is not conclusive from the data, since the crystallite sizes are essentially the same for all cases except 0.16M.

6. Lines 174-176 saying "appears to have an impact" is not scientific. Clear explanation mut be added in the claim "regulates the development rate of the ageing period of solution and nucleation". It is the first mention of ageing period and nucleation, so these need to be supported with further evidence, for examples, references or additional data.

7. Section 2.3 Line 195 "intensity decreases" may not be clear unless additional details are added to how the FTIR analysis is done. Transmittance % depends on the film thickness, so kindly add details about the FTIR tool, measurement technique (e.g., transmission based or ATR), sample preparation details, and sample thickness. Why is there a shift in the base intensity 4000-2000cm-1 amongst the samples where no FTIR peak is present? Perhaps consider normalizing the data to the baseline, and then estimate the "peak intensity" changes.

8. Line 198 The change in the "peak position" in unclear. From Figure 4, it looks the same. Suggest adding a guiding line at those particular wavenumbers (similar to what was done in the XRD data).

9. Lines 255-260 Results suggest that precursor >0.16M must be studied to gain confidence in these results. Do we expect the crystallite size to keep decreasing? Also suggest doing repeats of the data to build statistical confidence in the rod size and CV data and to determine the variation in the values.

10. How do the powder cleaning techniques impact the rod sizes, or the CV curve? Is the difference in Cs values affected by cleaning process?

11. Line 263-264 The authors themselves recommend that more characterization is necessary to understand the structural and electrochemical properties of the ZnO structure. Kindly explain what the next steps should be and highlighted "future research directions" alluded to in Line 262.

12. Section 4.1 Line 273 "mixture solution is transferred into Teflon". Add details of the container, was it a Teflon beaker or a dish?

13. Section 4.2 Must add further details of the characterization tools used and the analysis procedure.

Author Response

Dear reviewer, 

Reviewer 3 Report

I have studied the manuscript entitled "Structural and electrochemical behaviors of ZnO structure: effect of different zinc precursor molarity" and provided some comments. Before further consideration in the journal of "Condensed Matter", the following comments need to be addressed:

1. Many reports are regularly published for ZnO-based materials nowadays wherein interesting aspects of the same are reported. Since ZnO-based materials have been extensively studied in the literature and due to the abundance of this subject the authors should emphasize their novelties concerning others in this field.

2. X-ray diffraction section (structural properties) should come in first to well identify the features of samples.

3. Why only three dominant planes were considered to calculate the P(hkl) (the relative peak intensity orientation) in Eq. 1?

4. How many crystal planes that you use to calculate the crystallite size? Please mention.

5. While calculating the D values for the samples, have the authors considered the contribution of instrument broadening? Contributions to β due to the instrument and sample.

6. The authors are encouraged to extend and improve the structural studies section by calculating some essential parameters like lattice parameters (a and c), atomic packing fraction, and dislocation density.

7. From the SEM images, it is really hard to estimate the grain size of the prepared samples. Could the authors demonstrate SEM images at closer magnification? Besides, a size distribution histogram, using Digimizer software will be helpful for this purpose.

8. Numbering in equation 3 is missing. Please put the numbering.

9. Please provide a larger view of the FTIR spectra in the range of 400-1000 cm-1.

10. The conclusions of the work don’t convince me and it should be rewritten. It's short. The authors should clarify their findings in this section. Please state the most valuable sample among the four that gave the highest/good performance. Besides, the conclusion must be in the past tense.

11. There are some grammatical, typos, and technical errors. besides a few unclear sentence structures that exist in the whole manuscript. Please proofread them before more consideration. Some of them are listed as follows:

"…of reactants that classified into two types" should be replaced with "…of reactants that were classified into two types"

"litium" by "lithium"

"Various shape" by "Various shapes"

"Zinc oxide crystallite and pore diameters that ranges were determined" should be replaced

with "Zinc oxide crystallite and pore diameter ranges were determined"

"followed by decreasing of the resistivity" by "followed by decreasing in resistivity"

"we focused to synthesis the ZnO structure by using" should be replaced

with "we focused on the synthesis of the ZnO structure by using"

"ZnO samples at different Zn precursor solution" by "ZnO samples in different Zn precursor

solutions"

"aspect ratio increases and then enhancing their chemical" by "aspect ratio increases and then

enhances their chemical"

"distribution seem not so good due to compact structure is" by "distribution seems not so good

due to the compact structure being"

"fourier transform infrared (FTIR)" by "Fourier transform infrared (FTIR)"

"In summary, the ZnO structure samples was successfully" by "In summary, the ZnO structure samples were successfully"

"wurzite" by "wurtzite"

And some others…

Author Response

Dear Reviewer,

Reviewer 4 Report

Mohamed et al. reported the structural and electrochemical evolutions of ZnO nanoparticles via changing the ZnO molarity and investigated the properties by SEM, XRD, CV, etc. The referee would suggest paying attention to the following comments before having a decision on publication. These comments are arranged in order of appearance rather than in order of significance.

In the abstract, the authors claim that “SEM images showed that ZnO samples were in hexagonal structure.” This statement is not correct. In fig 1, these blurred SEM images show a rectangular shape in addition to the hexagonal structures, then SEM is not used to solve the crystal structure. The expression should be precise.

 The crystal lattice parameters usually have an inverse relation with the peak position. In fig.2, why the crystal size increases from 0.1 M to 0.14M and decreases again in 0.16M? Is the zero shift in the XRD measurement considered?

A lot of typos and grammar errors need to be corrected. For instance, “litium” should be lithium in Line 45 on Page 2. “have” in line 90 on Page 2 should be “has”.

Author Response

Dear reviewer

Round 2

Reviewer 1 Report

I appreciate the attention given to all my comments and suggestions. The authors have made changes and improvements in their manuscript and also corrected some technical errors. They have given convincing responses to most of the questions and comments that I had raised.

The revised version of the manuscript appears technically sounder. The paper can be accepted for publication.

Reviewer 3 Report

After the authors' careful revision the manuscript seems fluent now. The authors have responded properly to the comments and suggestions. No further amendment is needed. The present form of the manuscript can be considered for publication.